# Na^+^/K^+^- and Mg^2+^-ATPases and Their Interaction with AMPA, NMDA and D_2_ Dopamine Receptors in an Animal Model of Febrile Seizures

**DOI:** 10.3390/ijms232314638

**Published:** 2022-11-24

**Authors:** María Crespo, David Agustín León-Navarro, Mairena Martín

**Affiliations:** 1Department of Inorganic, Organic Chemistry and Biochemistry, Faculty of Chemical and Technological Sciences, Regional Centre of Biomedical Research (CRIB), Universidad de Castilla-La Mancha, 13071 Ciudad Real, Spain; 2Department of Inorganic, Organic Chemistry and Biochemistry, Faculty of Chemical and Technological Sciences, School of Medicine of Ciudad Real, Regional Centre of Biomedical Research (CRIB), Universidad de Castilla-La Mancha, 13071 Ciudad Real, Spain

**Keywords:** Na^+^/K^+^ ATPase, Mg^2+^ ATPase, AMPA receptor, NMDA receptor, febrile seizure, epilepsy

## Abstract

Febrile seizures (FS) are one of the most common seizure disorders in childhood which are classified into short and prolonged, depending on their duration. Short FS are usually considered as benign. However, epidemiological studies have shown an association between prolonged FS and temporal lobe epilepsy. The development of animal models of FS has been very useful to investigate the mechanisms and the consequences of FS. One of the most used, the “hair dryer model”, has revealed that prolonged FS may lead to temporal lobe epilepsy by altering neuronal function. Several pieces of evidence suggest that Na^+^/ K^+^-ATPase and Mg^2+^-ATPase may play a role in this epileptogenic process. In this work, we found that hyperthermia-induced seizures (HIS) significantly increased the activity of Na^+^/ K^+^-ATPase and Mg^2+^-ATPase five and twenty days after hyperthermic insult, respectively. These effects were diminished in response to AMPA, D_2_ dopamine A_1_ and A_2A_ receptors activation, respectively. Furthermore, HIS also significantly increased the protein level of the AMPA subunit GluR1. Altogether, the increased Na^+^/ K^+^-ATPase and Mg^2+^-ATPase agree well with the presence of protective mechanisms. However, the reduction in ATPase activities in the presence of NMDA and AMPA suggest an increased propensity for epileptic events in adults.

## 1. Introduction

Febrile seizures (FS) are one of the most common seizure disorders in childhood between 6 months and 5 years, affecting about 2–4% of children. FS are brought on by fever but without evidence of infection of the central nervous system [1,2]. FS can be classified into two main types according to their duration: short FS, which last less than 15 min, and prolonged FS, which last longer than 15 min. Generally, short FS are considered benign. However, the outcome of prolonged FS is more controversial. Thus, whereas retrospective studies have shown an association between prolonged FS and temporal lobe epilepsy, prospective studies have failed to find such a relationship [3,4].

The development of animal models of FS has been very useful to investigate their mechanisms and consequences. One of the most used is the “hair dryer model”, in which hyperthermia-induced seizures (HIS) are evoked in neonatal rats by an adjustable stream of heated air [5]. This animal model has revealed that prolonged fever seizures lead to temporal lobe epilepsy in 30% of rats [6]. Moreover, although the mechanisms by which prolonged FS could contribute to epilepsy remains unknown, no cell death and neuronal circuit reorganization seems to be involved, suggesting changes in neuronal function as the main mechanism in the epileptogenic process during development [4,7,8]. In that sense, multiple works carried out in recent years have revealed that HIS evoke changes in metabotropic receptors, such as adenosine A_1_ and A_2A_ receptors [9,10,11,12,13], metabotropic glutamate receptor [14] or GABA_B_ receptor [15]. Ionotropic receptors, including NMDA and GABA_A_ receptors, are also modulated by FS [16,17,18,19].

Several pieces of evidence suggest that the Na^+^/ K^+^-ATPase may play a key role in the epileptogenic process. Thus, fundamental cellular processes commonly associated to seizure episodes, such as neuronal excitability or neurotransmitter release, are regulated by Na^+^/ K^+^-ATPase activity. Moreover, multiple works using epileptic animal models or human brain have found mutations in the Na^+^/ K^+^-ATPase α subunit gene [20,21]. Moreover, changes in Na^+^/ K^+^-ATPase activity have also been found in rodent brain after pentylentetrazol-induced seizures, and in epileptic human brain [22,23]. On the other hand, a variety of studies have shown that Na^+^/ K^+^-ATPase activity can be modulated by neurotransmitter receptors; this process seems to involve a direct interaction in the case of glutamate ionotropic receptor NMDA and AMPA, forming macrocomplexes for NMDA [24] or binding directly to the Na^+^/ K^+^-ATPase subunit alpha in the case of AMPA [25]. On the other hand, metabotropic receptors seem to involve mechanisms mediated by protein kinases activities through secondary messenger signaling cascades [26,27].

The role of Mg^2+^-ATPase in the epileptogenic process is less known. This pump contributes to maintaining a high intracellular concentration of magnesium. Since these ions are crucial for several cellular processes such as enzymatic reactions, cellular signaling and ion channel functions [28] it seems plausible that Mg^2+^-ATPase could be altered in response to HIS.

In the present work, we analyzed the activity of both Na^+^/ K^+^ and Mg^2+^-ATPases in cortical plasma membranes from rat brain at five and twenty days after HIS and at 2-months-old. Furthermore, we studied the status of AMPA, NMDA and dopamine receptor and the effect of their activation on Na^+^/ K^+^-ATPase and Mg^2+^-ATPase activities.

## 2. Results

### 2.1. Na^+^/K^+^-ATPase Activity in Cortical Plasma Membranes after HIS

To know whether HIS can alter Na^+^/K^+^-ATPase activity, we measured the enzymatic activity at 5 and 20 days after hyperthermic insult and in 2-month-old animals. As shown in Figure 1, no significant variation was detected in Na^+^/K^+^-ATPase activity 5 days after HIS (439.8 ± 62.79 nmol of Pi /mg prot /min vs. 644.5 ± 78.74 nmol of Pi /mg prot /min, *p* > 0.05). However, a significant increase was observed 20 days after hyperthermic seizures (73.7 ± 8.22 nmol of Pi /mg prot /min vs. 143.5 ± 14.08 nmol of Pi /mg prot /min, *p* < 0.05) and in 2-month-old animals after hyperthermic seizures (24 ± 4 nmol of Pi /mg prot /min vs. 50 ± 9 nmol of Pi /mg prot /min, *p* < 0.05) (Figure 1).

### 2.2. Modulation of Na^+^/K^+^-ATPase Activity by G-Protein-Coupled Receptor after HIS

Next, we analyzed whether the activation of different G-protein-coupled receptors could modify the increased Na^+^/K^+^-ATPase activity observed 20 days after HIS. Thus, we assayed three agonists: sumanirole, CPA and GGS 21680, which act through dopamine D_2_ and adenosine A_1_ and A_2A_ receptors, respectively. A two-way ANOVA was carried out to analyze the effect of HIS and agonists on Na^+^/K^+^-ATPase activity. This analysis revealed that there was a significant effect of sumanirole [F(1,14) = 20.73, *p* = 0.0005], HIS [(F(1,14) = 15.11, *p* = 0.0016] and sumanirole x HIS interaction [F(1,14) = 6.74, *p* = 0.0211] on Na^+^/K^+^-ATPase activity (Figure 2A). The Bonferroni’s post test showed that HIS induced a significant increase in the activity of Na^+^/K^+^-ATPase in control rats, and that this effect was significantly reduced by sumanirole (Figure 2A).

With respect to CPA, a selective A_1_ receptor agonist, the two-way ANOVA analysis revealed that HIS and agonist significantly altered the activity of Na^+^/K^+^-ATPase [HIS: F(1,15) = 18.37, *p* = 0.0006; CPA: F(1.15) = 6.517, *p* = 0.0221] without observed HIS x agonist interaction (Figure 2B). Similar results were obtained when CGS 21680, selective A_2A_ receptor agonist, was assayed. [HIS: F(1,15) = 16.8, *p* = 0.0009; CGS 21680: F(1.15) = 4.58, *p* = 0.0492] (Figure 2C). Bonferroni’s post-test only showed an increase in Na^+^/K^+^-ATPase activity in the control group after HIS for all agonists assayed.

### 2.3. Modulation of Na^+^/K^+^-ATPase Activity by Ionotropic Glutamate Receptor after HIS

To evaluate the effect of ionotropic glutamate receptor, we used AMPA and NMDA, selective agonists of the different types of the corresponding ionotropic receptors. The two-way ANOVA analyses revealed that both HIS and agonists [NMDA: F(1,14) = 4.729, *p* = 0.0473 and AMPA: F(1,14) = 26.56, *p* = 0.0001] significantly altered Na^+^/K^+^-ATPase activity. No significant interaction of HIS x agonist was observed for NMDA and AMPA receptor (Figure 3A,B). Bonferroni’s post hoc test showed that the significant increase in Na^+^/K^+^-ATPase activity induced by HIS was significantly reduced in the presence of AMPA (*p* < 0.01) (Figure 3B).

### 2.4. Modulation of Na^+^/K+-ATPase Activity after HIS by AMPA and Sumanirole in 2-Month-Old Animals

The effect of ionotropic glutamate receptor AMPA and G-protein-coupled receptor dopamine D_2_ receptor was shown in the Na^+^/K^+^-ATPase activity. The two-way ANOVA analyses revealed a significant difference in HIS and agonist AMPA [AMPA: F (1,12) = 17.97, *p* = 0.0011] (Figure 4A) in Na^+^/K^+^-ATPase activity. However, no significative interaction of HIS x agonist was found.

Bonferroni’s post hoc test showed a significant increase in Na^+^/K^+^-ATPase activity after HIS in the control group in Figure 4A and B (*p* < 0.01), with an increase in Na^+^/K^+^-ATPase activity in the sumanirole group in Figure 4B (*p* < 0.05). In Figure 4B, a decrease was observed in Na^+^/K^+^-ATPase activity after HIS in the presence of agonist (*p* < 0.05).

### 2.5. Mg^2+^-ATPase Activity in Cortical Plasma Membrane after HIS

The analysis of Mg^2+^-ATPase activity revealed that it was significantly increased 5 days after HIS (175.4 ± 12.94 nmol of Pi /mg prot /min vs. 270.9 ± 7.98 nmol of Pi /mg prot /min, *p* < 0.01), whereas no significant variation was found 20 days following hyperthermic seizures (136.1 ± 13.91 nmol of Pi /mg prot /min vs. 131.7 ± 20.46 nmol of Pi /mg prot /min, *p* > 0.05) and in 2-month-old animals (134.0 ± 10.88 nmol of Pi /mg prot /min vs. 126.1 ± 8.40 nmol of Pi /mg prot /min, *p* > 0.05) (Figure 5).

### 2.6. Modulation of Mg^2+^-ATPase Activity by G-Protein-Coupled Receptor after HIS

We also analyzed whether the increase in Mg^2+^-ATPase activity induced by HIS could be modulated by sumanirole, CPA or CGS 21680. The two-way ANOVA analyses showed that HIS caused a significant change in Mg^2+^-ATPase activity [Figure 6A: F(1.16) = 18.23, *p* = 0.0006; Figure 6B: F(1.16) = 15.28, *p* = 0.0013; Figure 6C: F(1.16) = 14.7, *p* = 0.0115]; moreover, agonist x HIS interaction was detected for CPA and CGS 21680 agonist [CPA: F(1.16) = 6.452, *p* = 0.0218; CPA x HIS: F(1.16) = 5.604, *p* = 0.0309; CGS 21680: F(1.16) = 8.037, *p* = 0.0119; CGS 21680 x HIS: F(1.16) = 5.475, *p* = 0.0326].

Bonferroni’s post hoc test showed an increase in Mg^2+^-ATPase activity for sumanirole after HIS in the control group (*p* < 0.05), while in the presence of CPA and CGS 21680 we observed an increase in Mg^2+^-ATPase activity after HIS in the control group (*p* < 0.01), together with a decrease in this activity after HIS in the presence of both agonists (*p* < 0.05).

### 2.7. Modulation of Mg^2+^-ATPase by Ionotropic Receptors after HIS

Finally, we analyzed the effect of ionotropic glutamate receptors activation on the increase in Mg^2+^-ATPase activity evoked by HIS. The two-way ANOVA revealed a significant effect of HIS [Figure 7A F(1,16) = 8.866, *p* = 0.0089; and Figure 7B F(1,16) = 20.58, *p* = 0.0003] on Mg^2+^-ATPase activity, similarly for NMDA, agonist and agonist x HIS [NMDA: F(1.16) = 7.788, *p* = 0.0131; NMDA x HIS: F(1.16) = 6.937, *p* = 0.0181] (Figure 7A).

Bonferroni’s post hoc test showed an increase in Mg^2+^-ATPase activity after HIS in the control group (*p* < 0.01). Moreover, Mg^2+^-ATPase activity was increased in the presence of AMPA (AMPA: *p* < 0.05). However, a significative decrease was observed after HIS with NMDA (*p* < 0.01).

### 2.8. Protein Level of Na^+^/K^+^ATPase and Receptors after HIS

The effect of HIS on the expression of the alpha subunit of Na^+^/K^+^-ATPase, dopamine D_2_ receptors and the subunits NR1 and GluR1 of NMDA and AMPA receptors was studied by a Western-blot analysis. As shown in Figure 8A,B, Figure 9A and Figure 10, no significant variation of the alpha subunit Na^+^/K^+^-ATPase, dopamine D_2_ receptor and NR_1_ subunit was found immediately after HIS, nor 5 or 20 days after. However, a significant increase in the subunit GluR1 of AMPA receptor was detected after 20 days, whereas it was unaltered 5 days after HIS (0.62 ± 0.06 vs. 1.21 ± 0.13, *p* < 0.01) (Figure 9B).

## 3. Discussion

The results obtained in the present work show that Na^+^/K^+^-ATPase activity significantly increased 20 days after HIS (PD 32) and in 2-month-old animals (PD 60). This effect was significantly reduced after activation of D_2_-dopamine and AMPA receptors. When these agonists were studied in 2-month-old animals, only AMPA showed a decrease in Na^+^/K^+^-ATPase activity. Furthermore, a significant increase in Mg^2+^-ATPase activity could be observed 5 days after HIS (PD 17), which was also reduced upon CPA, CGS 21680 and NMDA receptor activation.

Multiple works using animal models of epilepsy have shown changes in Na^+^/K^+^-ATPase activity, although the results are discrepant [23,29,30]. Our results agree well with those previously published by Reime Kinjo and coworkers [30], who described that three episodes of pilocarpine-induced status epilepticus at postnatal days 7, 8 and 9 evoked a significant increase in Na^+^/K^+^-ATPase activity in the hippocampus 7 or 30 days later. We believe that the increased Na^+^/K^+^-ATPase activity observed in cortical plasma membranes 20 days after HIS and maintained until 2 months of age could constitute a compensatory mechanism to reduce the extracellular concentration of potassium and, therefore, reduce the hyperexcitability of the cortex. Supporting this hypothesis, a previous study suggested that the convulsions in febrile seizures are evoked by a malfunction of temperature dependent Kir4.1 channel in astrocytes. This channel, under normal conditions, effectively removes extracellular potassium ions during the neuronal firing process. However, this channel reduced its amplitude at 40º C, being unable to remove extracellular excessive potassium ions, which lead to rapid accumulation and, consequently, the appearance of febrile seizures [31].

Another interesting result obtained in the present work is that the activation of AMPA and D_2_ dopamine receptors reduced the increase in Na^+^/K^+^-ATPase activity evoked by HIS 20 days after, and the AMPA effect was maintained over time up to 2 months of age. However, under control conditions, the activation of these receptors did not induce any significant effect on Na^+^/K^+^-ATPase activity. These results can be explained by the ability of Na^+^/K^+^-ATPase to associate physically to AMPA and D_2_ dopamine receptors. The colocalization of Na^+^/K^+^-ATPase and AMPA receptor has been shown in cultured cortical neurons, which was also accompanied by a specific association between the alpha subunit of Na^+^/K^+^-ATPase and AMPA receptors using coimmunoprecipitation in rat cortex lysates [25]. Finally, Hazelwood and coworkers [32], using coimmunoprecipitation and mass spectrometry, showed that in the brain, D_1_ and D_2_ dopamine receptors can form a complex with the Na^+^/K^+^-ATPase pump. Furthermore, they suggested that the D_2_-mediated decrease in Na^+^/K^+^-ATPase activity was dependent upon protein–protein interactions, rather than signaling molecules. Therefore, the results obtained in the present work suggest that HIS could promote a physical interaction between Na^+^/K^+^-ATPase and AMPA and D_2_ dopamine receptor. Taking into account that the increase in Na^+^/K^+^-ATPase activity detected 20 days after HIS and maintained at 2 months old for AMPA receptor constitute a compensatory mechanism to counterbalance the hyperexcitability of the cortex. The activation of these receptors diminished the efficacy of this mechanism and favored the appearance of epileptic status in adulthood.

The changes observed in the activity of Na^+^/K^+^-ATPase 20 days after HIS seems not to be associated to the corresponding increase in the protein level of this enzyme in the plasma membrane, since the densitometric analyses of the bands in the Western blot experiments failed to find any significant variation. A similar lack of significant variation was also observed when the protein levels of D_2_ dopamine receptor and NR1 subunit of NMDA receptors were studied. Taking into account that NMDA receptors are formed by three different subunits (NR1, NR2 and NR3) and that NR1 is essential for channel formation [33], the results obtained in this work suggest that HIS did not significantly modify the protein level of NMDA in cortical plasma membranes. However, the immunochemical detection of the subunit GluR_1_ revealed that HIS induced a significant increase in this subunit 20 days after hyperthermic seizures. The AMPA receptor is a tetramer composed of four subunits of GluR1, GluR2, GluR3 and GluR4. An increase in GluR1 subunit’s protein has previously been shown in the hippocampus of epileptic patients [34]. A similar increase in GluR1 has also recently been published in an immature rodent model of temporal lobe epilepsy [35], where the intraperitoneal injection of Kainate in three-week-old mice evoked seizures and an increase in the level of GluR1 expression. Therefore, the increase in GluR1 subunits observed in the present work seems to favor the development of an epileptogenic process.

In 2-month-old animals, we found that D_2_ dopamine receptor remained unaltered, as at 20 days after HIS. However, GluR_1_ levels, analyzed for AMPA receptor, that were increased 20 days after HIS, recovered the control values. Therefore, the relationship between Na^+^/K^+^-ATPase and AMPA does not seem to depend on the levels of AMPA in the membrane in that case.

Concerning Mg^2+^-ATPase activity, the results show that HIS significantly increased the activity of this enzyme 5 days after HIS. This enzyme maintains high intracellular concentrations of magnesium, which is important to regulate several biochemical reactions involved in energy metabolism, signal transduction and protein synthesis [36,37]. Thus, magnesium plays a key role in the intracellular signaling process by encouraging protein kinase activity [38]. Moreover, magnesium provides protective effects against free radicals and contributes to genome stability by acting as a cofactor for enzymes involved in DNA repair [39]. Therefore, the increase in Mg^2+^-ATPase activity observed in the present work suggests the existence of a protective mechanism. Accordingly, de Freitas and coworkers [40] found an increase in Mg^2+^-ATPase activity in rat hippocampus after pilocarpine-induced seizures, which was attributed to the oxidative stress caused by the seizures. In that sense, previous work carried out in our laboratory has shown that HIS evokes oxidative stress in the cerebellum and cortex of rats [10,41].

Finally, the results in this work show that HIS promotes a negative interaction between NMDA, A_1_ and A_2A_ receptor and Mg^2+^-ATPase activity. Thus, the increase in Mg^2+^-ATPase activity detected in the hyperthermia group was significantly reduced after NMDA, A_1_ and A_2A_ receptor activation. There are no reports, at least to our knowledge, showing such an interaction. Considering that the increase in Mg^2+^-ATPase activity following HIS is a protective mechanism, the reduction in this effect could increase the risk of suffering epileptic disorder in adulthood. More work would be helpful to clarify this point.

In summary, after hyperthermia-induced seizures at postnatal day 12, a significant increase in Na^+^/K^+^-ATPase 20 days and 2 months later, respectively, and Mg^2+^-ATPase activity 5 days later, was observed, which agrees well with the existence of a protective mechanism. However, these effects were also accompanied by an increase in the AMPA subunit GluR1 and a negative interaction between Na^+^/K^+^-ATPase and Mg^2+^-ATPase and AMPA and NMDA receptor, respectively, which seem to indicate the loss of such neuroprotective mechanisms increasing the vulnerability and propensity to epileptic events in adults.

## 4. Materials and Methods

### 4.1. Materials

N6-cyclopentyladenosine (CPA), adenosine triphosphate (ATP) and malachite green were from Sigma-Aldrich (Madrid, Spain). Ammonium molybdate was from Merck (Madrid, Spain). CGS 21680, NMDA, AMPA and Kainate were from Tocris Bioscience (Madrid, Spain). All other reagents were of analytical grade and obtained from commercial sources.

### 4.2. Animals

Pregnant rats were obtained from the Autonomous University of Madrid (Spain). The care and use of animals were carried out in accordance with the Spanish laws (RD 53/2013 and Ley 32/2007) governing the use of laboratory animals. All experiments were performed under the guidelines of the Animal Experimental Committee of the University of Castilla-La Mancha. Every effort was made to minimize animal suffering and to reduce the number of animals used. Animals were maintained in a 12 h light/12 h dark cycle (lights on at 07:00 h) in a temperature-controlled room (25 °C) with constant humidity (40–50%) and with free access to food and drinking water. The day of birth was considered day 0. Pups were housed with their mother until weaning at postnatal day (PD) 21.

### 4.3. Hyperthermia-Induced Seizures

Neonatal rats were submitted to hyperthermia-induced seizures (HIS) at PD 12. This developmental stage is equivalent to a human aged between several months and 3 years, a period especially susceptible to febrile seizures. Hyperthermia was induced using a warmed air stream (45–50 °C) from a hair dryer located 50 cm above a plastic chamber (17 × 12 × 12 cm), using a previously described protocol [10,11]. Rectal temperature was measured at a 2 min interval. In order to judge the occurrence of seizures, two observers monitored the rats’ behavior. The behavioral seizures were stereotyped and previously shown to correlate with electroencephalography (EEG) discharges [5]. These behavioral seizures consisted of the arrest of heat-induced hyperkinesia, followed by body flexion. All pups exhibited these behaviors as well as rearing and falling over associated with hind-limb clonus seizures (stage 5 on Racine scale criteria), at ~10 min after hyperthermia induction when the core temperature was ~42 °C. After 20 min of behavioral seizures, the pups were moved to a cool surface until their rectal temperature reached baseline, and they were then returned to the home cages with their mothers. The duration of heating was 27 ± 3 min. Controls were separated from their mothers for the same duration and placed in a chamber at room temperature.

### 4.4. Cerebral Cortical Membrane Isolation

Cerebral cortical membranes were isolated following the previously described protocol [42]. Briefly, the cortex brain from pups was homogenized in isolation buffer (50 mM of Tris-HCl, pH 7.4 containing 10 mM of MgCl_2_ and protease inhibitors) in a Dounce homogenizer. After homogenization, the samples were centrifuged for 5 min at 1000× *g* in a Beckman JA 21 centrifuge (Coulter, Madrid, Spain). Then, supernatant was centrifuged for 10 min at 27,000× *g*, and the pellet was finally resuspended in isolation buffer. Protein concentration was measured with the Lowry method, using bovine serum albumin as a standard.

### 4.5. Na^+^/K^+^-ATPase and Mg^2+^-ATPase Activities Assay

Enzyme activities were analyzed following the method previously described by Schweinberger and coworkers with minor modifications [43].

The reaction mixture contained 5.0 mM of MgCl_2_, 80.0 mM of NaCl, 20.0 mM of KCl and 40.0 mM of Tris–HCl, pH 7.4, in a final volume of 200 μL. The reaction was initiated by the addition of 3 mM of ATP. The Mg^2+^-ATPase activity was measured with the addiction of 1.0 mM of ouabain. The Na^+^/K^+^-ATPase activity was measured as the difference between the total activity and the activity of the Mg^2+^-ATPase. The specific activity of the enzyme was expressed as nmol of Pi released per min per mg of protein.

The effect of AMPA, NMDA, Kainate, dopamine D_2_ and adenosine A_1_ and A_2A_ receptor activation receptor on these enzymes was measured following the same protocol, but with a 6 min preincubation with AMPA (140 µM), NMDA (140 µM), Kainate (140 µM), sumanirole (140 µM), CPA (140 µM) and CGS 21680 (140 µM) at 30 °C. For each sample, the percentage of action of each ligand on the basal activity of Na^+^/K^+^ and Mg^2+^-ATPase was measured.

### 4.6. Western-Blotting Assay

Plasma membranes samples (40 μg) were subjected to 7.5% polyacrylamide gel electrophoresis (PAGE) in the presence of sodium dodecyl sulphate (SDS). After electrophoretic separation, the proteins were transferred into nitrocellulose membranes using iBlot™ Dry Blotting System (Invitrogen, Barcelona, Spain) and blocked for 1 h with 5% nonfat- skimmed milk in phosphate-buffered saline. Immunodetection was carried out by incubating the nitrocellulose membranes with the corresponding primary antibody anti-alpha 1 Sodium Potassium ATPase (1:5000, ab7671 from Abcam), anti-NMDAR1 (1:1000, ab134308 from Abcam), recombinant Anti-Glutamate Receptor 1 (AMPA subtype) antibody [EPR19522] (1:1000, ab183797 from Abcam) and Anti-Dopamine D_2_ Receptor antibody (1:1000, ab88074 from Abcam). β-Actin antibody (1:5000, ab8226 from Abcam) and Ponceau red staining were used as a gel loading controls. After washing, the blots were incubated with horseradish peroxidase-coupled goat anti-rabbit or anti-mouse IgG (1:5000, GARPO 172–1019 and GAMPO 170–6516 from Bio-Rad). Bands were visualized using the ECL chemiluminescence detection kit from GE Healthcare (Madrid, Spain) in a G:Box chamber, and specific bands were quantified by densitometry using Gene-Tools software (Syngene). The Western blot membrane used to detect NMDAR1 was stripped using a stripping solution containing H_2_O_2_ 30% for 15 min at 37 °C. This method allows the inactivation (quenched) of horseradish peroxidase and, therefore, to label a second protein with a different antibody without any interference. After stripping, the solution membrane was reproved using Anti-Glutamate Receptor 1 (AMPA subtype) antibody.

### 4.7. Statistical and Data Analysis

Statistical comparisons were carried out using an unpaired two-tailed Student’s t-test and a two-way ANOVA, followed by a Bonferroni comparison post hoc test using GraphPad Prism 5. The results are expressed as mean ± standard error of the mean. Differences between mean values were considered statistically significant at *p* < 0.05.

## Figures and Tables

**Figure 1 ijms-23-14638-f001:**
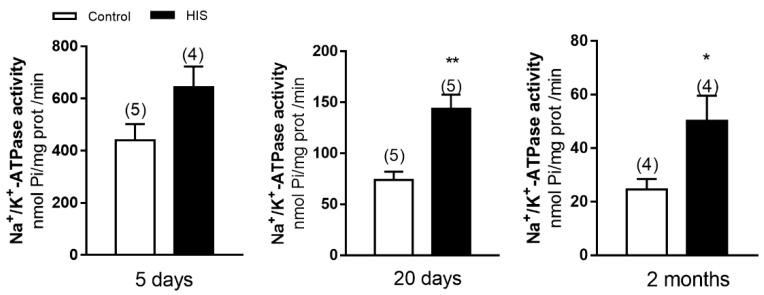
Status of Na^+^/K^+^-ATPase activity in neonatal cortex 5 and 20 days after HIS and in 2-month-old animals. Bar graphs show the effect of HIS on Na^+^/K^+^ ATPase activity measured 5 days and 20 days after HIS (PD 17 and PD 32, respectively) and in 2-month-old animals (PD60). Data are mean ± S.E.M. values from 4 to 5 different animals from independent litters. * *p* < 0.05 and ** *p* < 0.01 are significantly different from control using unpaired two-tailed Student’s *t*-test.

**Figure 2 ijms-23-14638-f002:**
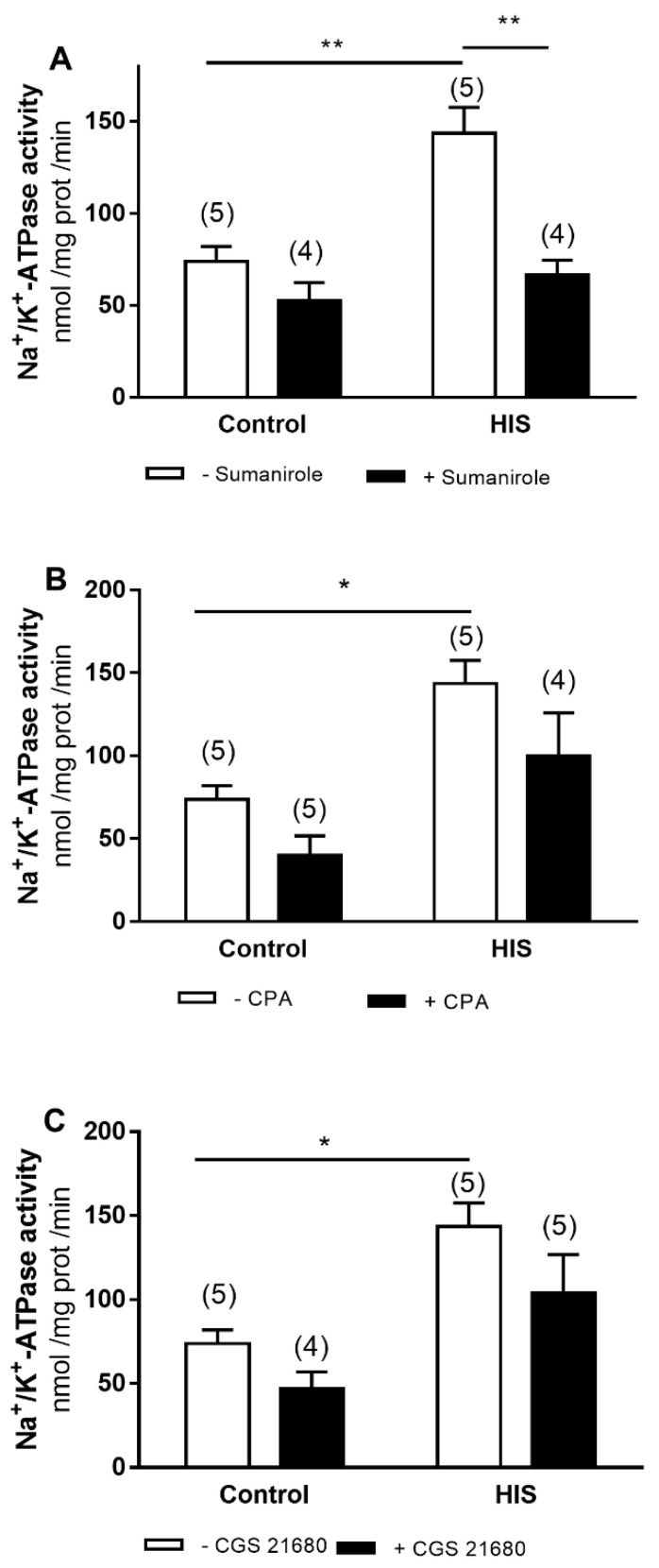
Effects of D_2_ dopamine (**A**), adenosine A_1_ (**B**) and A_2A_ receptors (**C**) activation on Na^+^/K^+^ ATPase activity in neonatal cortex 20 days after HIS (PD 32). The activity of Na^+^/K^+^ ATPase was measured in control rats (PD 32) and after HIS (PD 32), following the protocol described in the Materials and Methods section. To evaluate the influence of D_2_ dopamine, adenosine A_1_ and A_2A_ receptor activation on Na^+^/K^+^ ATPase activity, the corresponding plasma membranes were preincubated for 6 min at 30 °C with 140 μM of sumanirole, 140 μM of CPA and 140 μM of CGS 21680, respectively. Each bar represents the mean ± SEM for the indicated *n* value. * *p* < 0.05, ** *p* < 0.01 using a two-way ANOVA and Bonferroni’s post hoc test.

**Figure 3 ijms-23-14638-f003:**
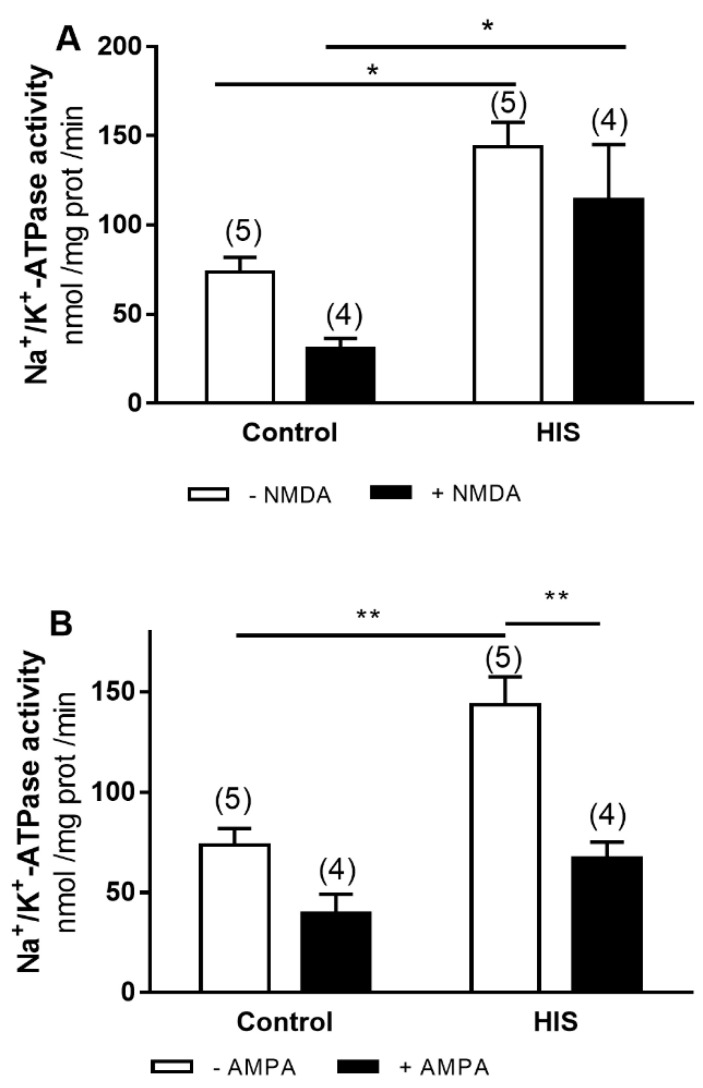
Effects of NMDA (**A**) and AMPA (**B**) activation on Na^+^/K^+^-ATPase activity in neonatal cortex 20 days after HIS (PD 32). The activity of Na^+^/K^+^ ATPase was measured in control rats (PD 32) and after HIS (PD 32), following the protocol described in the Materials and Methods section. To evaluate the influence of NMDA and AMPA receptor activation on Na^+^/K^+^ ATPase activity, the corresponding plasma membranes were preincubated for 6 min at 30 °C with 140 μM of NMDA and 140 μM of AMPA, respectively. Each bar represents the mean ± SEM for 4–5 values. * *p* < 0.05 and ** *p* < 0.01 using a two-way ANOVA and Bonferroni’s post hoc test.

**Figure 4 ijms-23-14638-f004:**
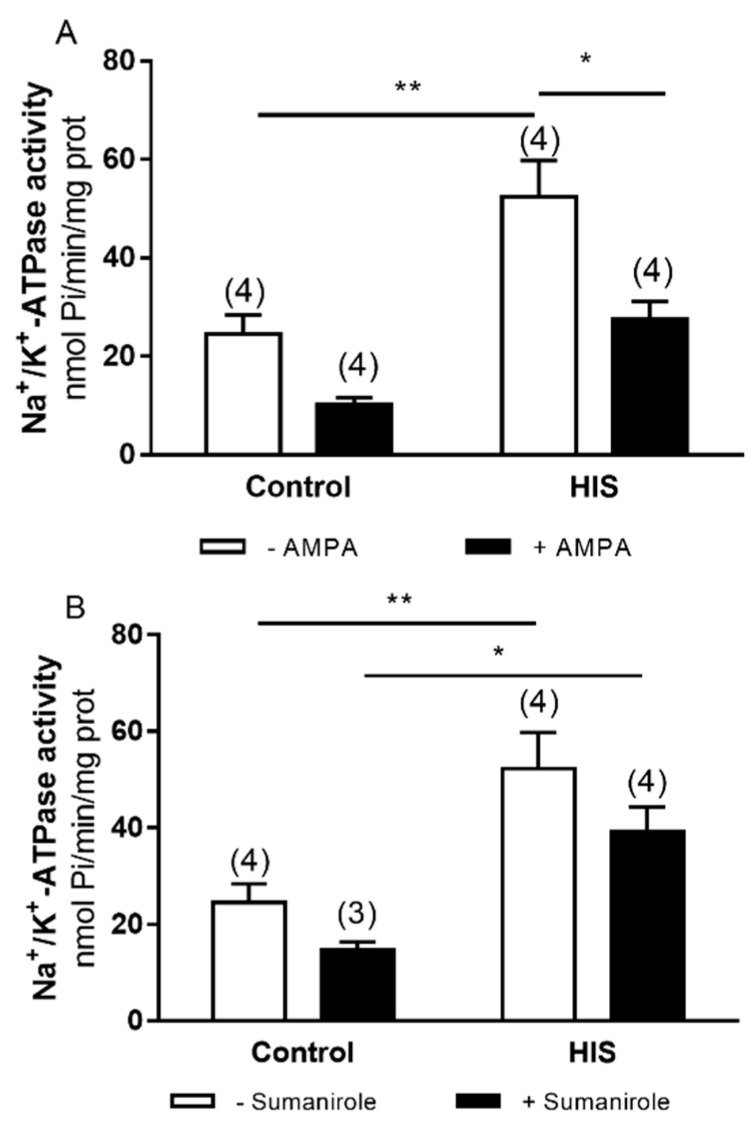
Effects of AMPA (**A**) and D_2_ dopamine (**B**) receptors activation on Na^+^/K^+^ ATPase activity in cortex after HIS in 2-month-old animals (PD 60). The activity of Na^+^/K^+^ ATPase was measured in control rats (PD 60) and after HIS (PD 60), following the protocol described in the Materials and Methods section. To evaluate the influence of AMPA and D_2_ dopamine receptor activation on Na^+^/K^+^ ATPase activity, the corresponding plasma membranes were preincubated for 6 min at 30 °C with 140 μM of AMPA and 140 μM of sumanirole, respectively. Each bar represents the mean ± SEM for for the indicated *n* value. * *p* < 0.05 and ** *p* < 0.01 using a two-way ANOVA and Bonferroni’s post hoc test.

**Figure 5 ijms-23-14638-f005:**
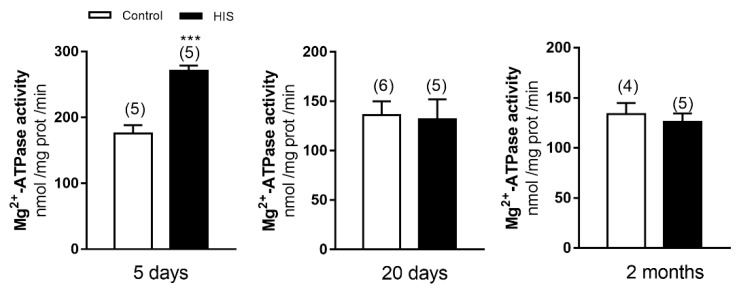
Status of Mg^2+^-ATPase activity in neonatal cortex 5 and 20 days after HIS and in 2-month-old animals. Bar graphs show the effect of HIS on Mg^2+^-ATPase activity measured 5 days and 20 days after HIS (PD 17 and PD 32, respectively) and in 2-month-old animals (PD 60). Data are mean ± SEM. values from 4 to 6 different neonates from independent litters. *** *p* < 0.001 is significantly different from control using unpaired two-tailed Student’s *t*-test.

**Figure 6 ijms-23-14638-f006:**
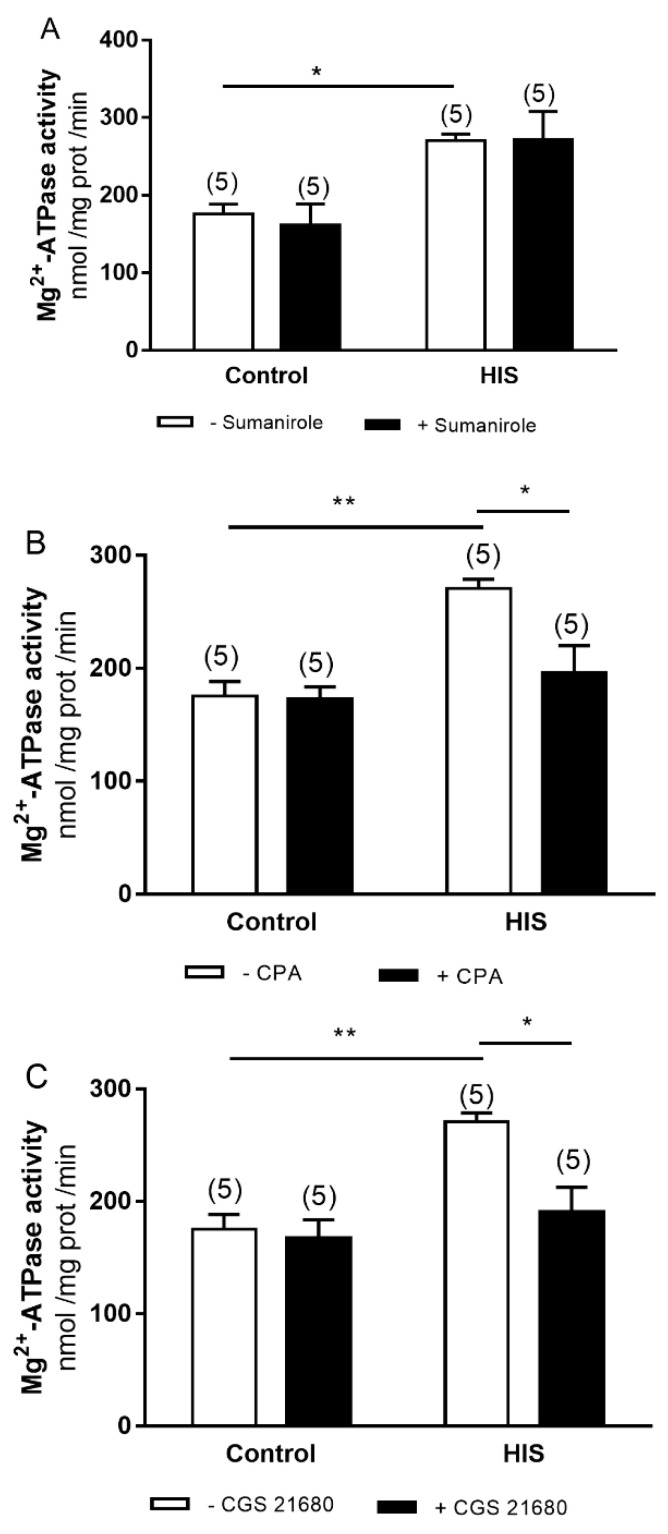
Effects of D_2_ dopamine (**A**), adenosine A_1_ (**B**) and A_2A_ receptors (**C**) activation on Mg^2+^-ATPase activity in neonatal cortex 5 days after HIS (PD 17). The activity of Mg^2+^-ATPase was measured in control rats (PD 17) and after HIS (PD 17), following the protocol described in the Materials and Methods section. To evaluate the influence of D_2_ dopamine, adenosine A_1_ and A_2A_ receptor activation on Na^+^/K^+^ ATPase activity, the corresponding plasma membranes were preincubated for 6 min at 30 °C with 140 μM of sumanirole, 140 μM of CPA and 140 μM of CGS 21680, respectively. Each bar represents the mean ± SEM for *n* = 5 in all groups analyzed. * *p* < 0.05 and ** *p* < 0.01 using a two-way ANOVA and Bonferroni’s post hoc test.

**Figure 7 ijms-23-14638-f007:**
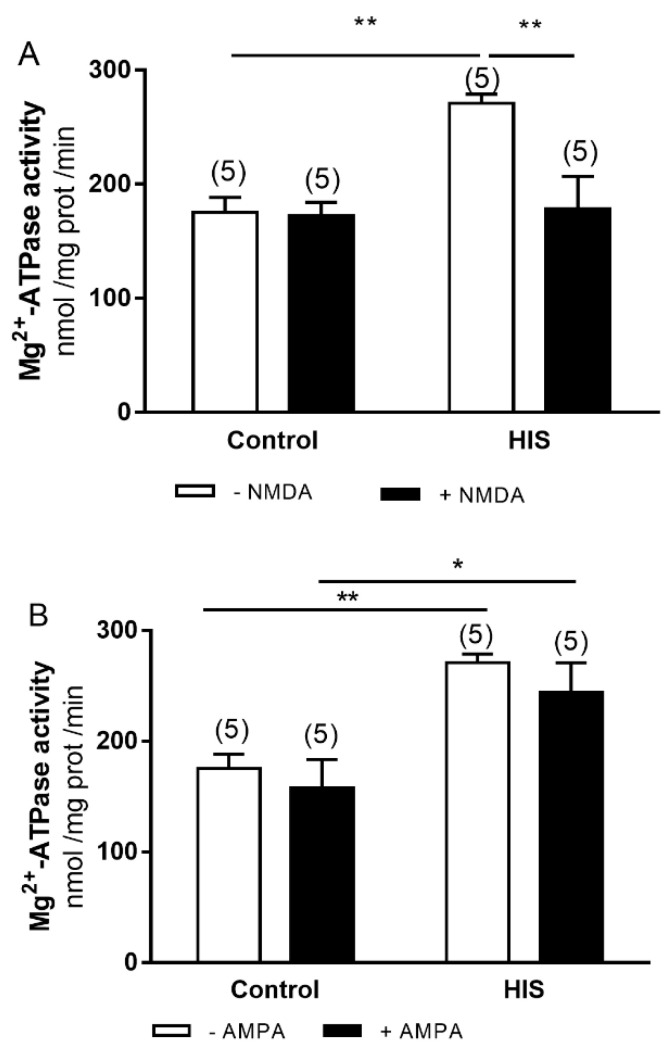
Effects of NMDA (**A**) and AMPA (**B**) receptors activation on Mg^2+^-ATPase activity in neonatal cortex 5 days after HIS (PD 17). The activity of Mg^2+^-ATPase was measured in control rats (PD 17) and after HIS (PD 17), following the protocol described in the Materials and Methods section. To evaluate the influence of NMDA and AMPA receptor activation on Mg^2+^-ATPase activity, the corresponding plasma membranes were preincubated for 6 min at 30 °C with 140 μM of NMDA and 140 μM of AMPA, respectively. Each bar represents the mean ± SEM for *n* = 5 * *p* < 0.05 and ** *p* < 0.01 using a two-way ANOVA and Bonferroni’s post hoc test.

**Figure 8 ijms-23-14638-f008:**
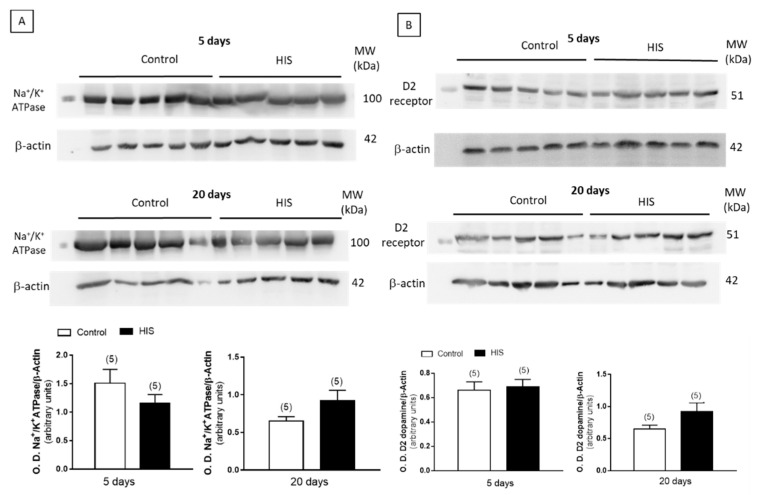
Immunoblotting analysis of alpha subunit of Na^+^/K^+^-ATPase (**A**) and dopamine D_2_ receptor (**B**) in cortical plasma membranes from rats 5 and 20 days after HIS. Plasma membranes (40 μg) were subjected to SDS-PAGE, electrophoretically transferred to nitrocellulose and probed with anti-alpha 1 Sodium Potassium ATPase and anti-Dopamine D_2_ receptor antibody antisera, as described in Materials and Methods. Data, expressed as arbitrary units, are means ± SEM of experiments performed with five different plasma membrane preparations (Appendix A).

**Figure 9 ijms-23-14638-f009:**
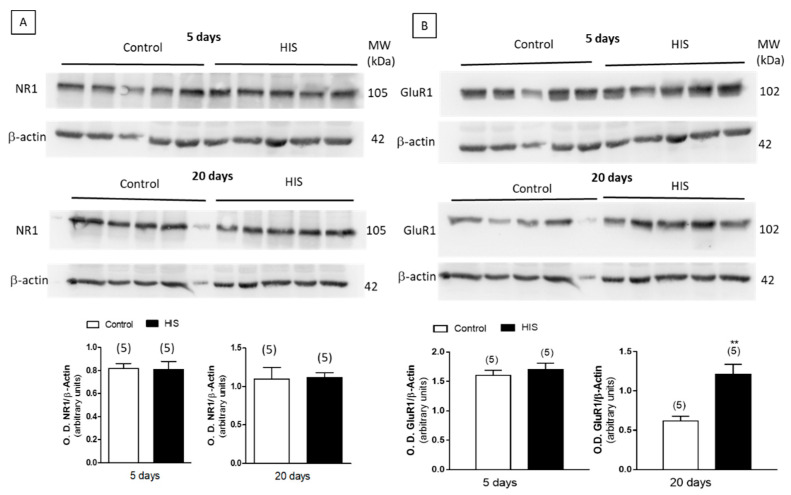
Immunoblotting analysis of alpha subunit of NR_1_ (**A**) and GluR_1_ (**B**) subunits in cortical plasma membranes from rats 5 and 20 days after HIS. Plasma membranes (40 μg) were subjected to SDS-PAGE, electrophoretically transferred to nitrocellulose and probed with anti-NMDAR1 and recombinant anti-Glutamate Receptor 1 antibody, as described in Materials and Methods. Data, expressed as arbitrary units, are means ± SEM of experiments performed with five different plasma membrane preparations. ** *p* < 0.01 using unpaired two tailed Student’s *t*-test (Appendix A).

**Figure 10 ijms-23-14638-f010:**
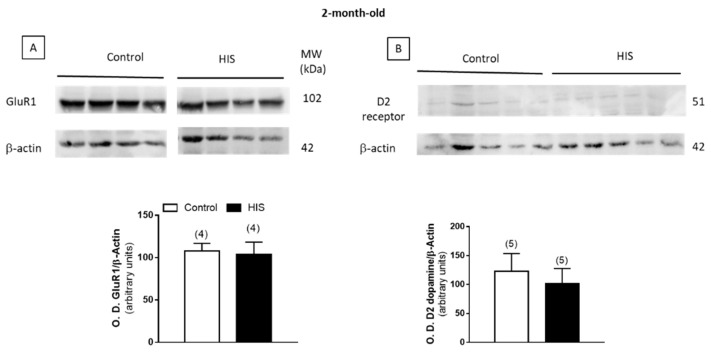
Immunoblotting analysis of alpha subunit of GluR1 (**A**) and dopamine D_2_ receptor (**B**) subunits in cortical plasma membranes from 2-month-old rats. Plasma membranes (40 μg) were subjected to SDS-PAGE, electrophoretically transferred to nitrocellulose and probed with recombinant anti-Glutamate Receptor 1 antibody and anti-Dopamine D_2_ receptor antibody antisera, as described in Materials and Methods. Data, expressed as arbitrary units, are means ± SEM of experiments performed with five different plasma membrane preparations (Appendix A).

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
