# Peer review of "Na+/K+- and Mg2+-ATPases and Their Interaction with AMPA, NMDA and D2 Dopamine Receptors in an Animal Model of Febrile Seizures"

_ijms, 2022, doi:10.3390/ijms232314638_

Round 1
Reviewer 1 Report
This sentence in the abstract has to be clarified.
“However, the reduction of both effects observed in response to AMPA and NMDA receptor activation suggest an increased vulnerability and propensity to epileptic events in adults.”
In the Results section, this part (lines 74-76) should be eliminated, because it is an editorial suggestion on how to write the paper…” This section may be divided by subheadings. It should provide a concise and precise description of the experimental results, their interpretation, as well as the experimental conclusions that can be drawn. “
You show changes in the activity of Na/K ATPase and other markers at 5 and 20 days after HIS (starting at P12 in rats), but what happens after months from HIS induction, it is not known. Since in the introduction you argument on the possible consequences of febrile seizures of children in adult subjects, it would be interesting to see what happens at a late stage of development in adult rats (>P60) at least as a future perspective of this study. Please, add new data or address better this issue.
Clearly, a limitation of this work is the absence of a study conducted by means of the electrophysiology, showing the functional correlate of this molecular with the presence of an hyperexcitable phenotype. Is it possible to add experiments done in this direction to corroborate the presence of FS in this model and what happens at different stages after the induction of HIS from the point of view of brain computation (so soon after 5days, later at 20 days and late in adult animals)? If not, please try to speculate more about this issue.
Author Response
ANSWERS TO REFEREE 1
Abstract sentence:
This sentence in the abstract has to be clarified.
“However, the reduction of both effects observed in response to AMPA and NMDA receptor activation suggest an increased vulnerability and propensity to epileptic events in adults.”
The sentence has been modified in order to a better understanding “However, the reduction in ATPase in the presence of AMPA and NMDA could suggest higher vulnerability to epileptic events in adults.”
In the Results section, this part (lines 74-76) should be eliminated, because it is an editorial suggestion on how to write the paper…” This section may be divided by subheadings. It should provide a concise and precise description of the experimental results, their interpretation, as well as the experimental conclusions that can be drawn.
We are so sorry by the mistake. Lines 74-76 has been deleted.
You show changes in the activity of Na/K ATPase and other markers at 5 and 20 days after HIS (starting at P12 in rats), but what happens after months from HIS induction, it is not known. Since in the introduction your argument on the possible consequences of febrile seizures of children in adult subjects, it would be interesting to see what happens at a late stage of development in adult rats (>P60) at least as a future perspective of this study. Please, add new data or address better this issue.
According to your suggestion, Na+/K+- and Mg2+-ATPases activities have been analysed in 2-month-old (PD 60) animals. These new results have been included in the revised version of the manuscript.
Clearly, a limitation of this work is the absence of a study conducted by means of the electrophysiology, showing the functional correlate of this molecular with the presence of an hyperexcitable phenotype. Is it possible to add experiments done in this direction to corroborate the presence of FS in this model and what happens at different stages after the induction of HIS from the point of view of brain computation (so soon after 5days, later at 20 days and late in adult animals)? If not, please try to speculate more about this issue.
The aim of the present work was to study possible molecular alterations in animals that have suffered febrile seizures early at postnatal period which is a risk factor for epilepsy in adulthood. We agree that electrophysiological study could to complete our results at molecular level. However, our group has no experience in electrophysiology and does not have the necessary infrastructure to be able to carry out electrophysiological studies.
Reviewer 2 Report
This is an interesting study showing the correlation between Na+ /K+ - and Mg2+-ATPases and AMPA, 2 NMDA and D2 dopamine receptors in an animal model of febrile seizures. However, most findings have been shown by other groups even with more mechanisms, for example how D1 and D2 dopamine receptors form complex with the Na+/K+-ATPase pump. The novelty of the study is insufficient. Secondly, only Na+ /K+ -ATPase and Mg2+-ATPase activities assay has been used in the whole study, more other ways of test are needed.
Author Response
ANSWERS TO REFEREE 2
This is an interesting study showing the correlation between Na+/K+ and Mg2+-ATPases and AMPA, 2 NMDA and D2 dopamine receptors in an animal model of febrile seizures. However, most findings have been shown by other groups even with more mechanisms, for example how D1 and D2 dopamine receptors form complex with the Na+/K+-ATPase pump. The novelty of the study is insufficient. Secondly, only Na+/K+-ATPase and Mg2+-ATPase activities assay has been used in the whole study, more other ways of test are needed
Previous results of our group in the same animal model of febrile seizures have showed alterations in adenosine and metabotropic glutamate receptors.
Crespo M, León-Navarro DA, Martín M. Glutamatergic System is Affected in Brain from an Hyperthermia-Induced Seizures Rat Model. Cell Mol Neurobiol. 2022 Jul;42(5):1501-1512. doi: 10.1007/s10571-021-01041-2. Epub 2021 Jan 25. PMID: 33492599.
Crespo M, León-Navarro DA, Martín M. Hyperthermia-induced seizures during neonatal period alter the functionality of A1 and A2A receptors in the cerebellum and evoke fine motor impairment and gait disturbances in adult rats. Physiol Behav. 2021 Oct 15;240:113543. doi: 10.1016/j.physbeh.2021.113543. Epub 2021 Jul 30. PMID: 34332977.
Crespo M, León-Navarro DA, Ruíz MÁ, Martín M. Hyperthermia-induced seizures produce long-term effects on the functionality of adenosine A1 receptor in rat cerebral cortex. Int J Dev Neurosci. 2020 Feb;80(1):1-12. doi: 10.1002/jdn.10000. Epub 2020 Jan 9. PMID: 31909494.
León-Navarro DA, Albasanz JL, Martín M. Hyperthermia-induced seizures alter adenosine A1 and A2A receptors and 5'-nucleotidase activity in rat cerebral cortex. J Neurochem. 2015 Aug;134(3):395-404. doi: 10.1111/jnc.13130. Epub 2015 May 13. PMID: 25907806.
In the present manuscript we try to analyses the Na+/K+ and Mg2+-ATPases which are involved in epilepsy. However, the revised version of the manuscript has been completed with new results from PD60 animals.
Reviewer 3 Report
In this work, Crespo et al., studied Na+/ K+-ATPase and Mg2+-ATPases activity in rat brain after hyperthermia-induced seizures (HIS). Moreover, the authors analyzed the effect of AMPA, NMDA and dopamine receptor activation on Na+/ K+-ATPase and Mg2+-ATPase activities.
Below my suggestions for the authors:
-
I suggest adding the values relative to single biological replicates over bar graphs;
-
I suggest to better describe in the introduction or in the discussion the effect of G protein; coupled and inotropic receptors on Na+/K+-ATPase and Mg2+-ATPase activity. This could help the reader in the understanding of the obtained results;
-
I suggest adding in the supplementary figures the name of the protein shown and the number of the figure of the main text to which they refer;
-
Please delete lines from 74 to 76 and from 376 to 382 (they refer to the template);
-
Lines 345-350: there is an unknown symbol.
Author Response
ANSWERS TO REFEREE 3
I suggest adding the values relative to single biological replicates over bar graphs
Each experiment was carried out in samples from different animals. Now, in the revised version of the manuscript, this valued was included over each bar graphs according to your suggestion
I suggest adding in the supplementary figures the name of the protein shown and the number of the figure of the main text to which they refer
The name of the protein shown and the number of figures which they refer has been included in the revised version of the manuscript.
I suggest to better describe in the introduction or in the discussion the effect of G protein; coupled and inotropic receptors on Na+/K+-ATPase and Mg2+-ATPase activity. This could help the reader in the understanding of the obtained results
We have added in the Introduction section a sentence to a better understanding the interaction between Na+/K+-ATPase and different receptors
“This process seems to involve a direct interaction in the case of NMDA and AMPA glutamate ionotropic receptor, forming macro complexes for NMDA [24] or directly binding to the Na+/ K+-ATPase subunit alpha in the case of AMPA [25]. On the other hand, metabotropic receptors seem to involve mechanisms mediated by protein kinases…”
Please delete lines from 74 to 76 and from 376 to 382 (they refer to the template)
These lines have been deleted.
Lines 345-350: there is an unknown symbol
This symbol has been corrected and changed for µ
Round 2
Reviewer 1 Report
I suggest to adjust bar hystograms with the same size/formattation (for example see figure 1 and 5 where the first two graphs are bigger than the second three graphs.
Reviewer 2 Report
The authors have provided proper additional info in the current version.